# Digital Pathology Applications for PD-L1 Scoring in Head and Neck Squamous Cell Carcinoma: A Challenging Series

**DOI:** 10.3390/jcm13051240

**Published:** 2024-02-22

**Authors:** Valentina Canini, Albino Eccher, Giulia d’Amati, Nicola Fusco, Fausto Maffini, Daniela Lepanto, Maurizio Martini, Giorgio Cazzaniga, Panagiotis Paliogiannis, Renato Lobrano, Vincenzo L’Imperio, Fabio Pagni

**Affiliations:** 1Department of Medicine and Surgery, Pathology, IRCCS Fondazione San Gerardo dei Tintori, University of Milano-Bicocca, 20126 Milan, Italy; v.canini@campus.unimib.it (V.C.); giorgio9cazzaniga@gmail.com (G.C.); vincenzo.limperio@unimib.it (V.L.); fabio.pagni@unimib.it (F.P.); 2Department of Medical and Surgical Sciences for Children and Adults, University of Modena and Reggio Emilia, University Hospital of Modena, 41124 Modena, Italy; 3Department of Radiological, Oncological and Pathological Sciences, Sapienza University of Roma, 00185 Rome, Italy; giulia.damati@uniroma1.it; 4Division of Pathology, European Institute of Oncology IRCCS, 20141 Milan, Italy; nicola.fusco@unimi.it (N.F.); fausto.maffini@ieo.it (F.M.); daniela.lepanto@ieo.it (D.L.); 5Department of Oncology and Hemato-Oncology, University of Milan, 20122 Milan, Italy; 6Department of Pathology, University of Messina, 98122 Messina, Italy; maurizio.martini@unime.it; 7Anatomic Pathology and Histology, Department of Medical, Surgical and Experimental Sciences, University of Sassari, 07100 Sassari, Italy; ppaliogiannis@uniss.it (P.P.); renato.lobrano@uniss.it (R.L.)

**Keywords:** PD-L1, digital pathology, head and neck squamous cell carcinoma, combined positive score

## Abstract

The assessment of programmed death-ligand 1 (PD-L1) combined positive scoring (CPS) in head and neck squamous cell carcinoma (HNSCC) is challenged by pre-analytical and inter-observer variabilities. An educational program to compare the diagnostic performances between local pathologists and a board of pathologists on 11 challenging cases from different Italian pathology centers stained with PD-L1 immunohistochemistry on a digital pathology platform is reported. A laboratory-developed test (LDT) using both 22C3 (Dako) and SP263 (Ventana) clones on Dako or Ventana platforms was compared with the companion diagnostic (CDx) Dako 22C3 pharm Dx assay. A computational approach was performed to assess possible correlations between stain features and pathologists’ visual assessments. Technical discordances were noted in five cases (LDT vs. CDx, 45%), due to an abnormal nuclear/cytoplasmic diaminobenzidine (DAB) stain in LDT (n = 2, 18%) and due to variation in terms of intensity, dirty background, and DAB droplets (n = 3, 27%). Interpretative discordances were noted in six cases (LDT vs. CDx, 54%). CPS remained unchanged, increased, or decreased from LDT to CDx in three (27%) cases, two (18%) cases, and one (9%) case, respectively, around relevant cutoffs (1 and 20, *k* = 0.63). Differences noted in DAB intensity/distribution using computational pathology partly explained the LDT vs. CDx differences in two cases (18%). Digital pathology may help in PD-L1 scoring, serving as a second opinion consultation platform in challenging cases. Computational and artificial intelligence tools will improve clinical decision-making and patient outcomes.

## 1. Introduction

Implementing immunohistochemical (IHC) testing for programmed death-ligand 1 (PD-L1) is a great field of interest for pathologists [1,2,3]. Several antibody clones were used in the context of “assays” associated with specific protocols, detection systems, and platforms, later approved by the Food and Drug Administration (FDA) as “companion diagnostics” (CDx) or as “complementary diagnostics” related to those specific immunotherapy drugs [1,3,4,5,6]. The combined positive score (CPS) is routinely required for head and neck squamous cell carcinomas (HNSCC) [7]. Various pre-analytical and interpretative factors can affect the test [1]. The CPS scoring system takes into account the expression of PD-L1 both by the tumor and inflammatory cells present in the intra and peritumoral stroma. In the first pembrolizumab registrative trial, the PD-L1 status was evaluated through an IHC staining performed on the Dako Autostainer Link 48 platform using the Dako 22C3 pharmDx assay, which was later approved by the FDA as a CDx [7]. The European Medicine Agency (EMA) instead approved its use without indicating the need for a specific test (laboratory-developed test, LDT). However, an IHC assessment of PD-L1 requires both the appropriate validation of the staining method and an evaluation by trained/experienced pathologists in a standardized and reproducible way. In this study, we report the results of an educational program for pathologists from all over Italy (Rome, May 2023) aimed at investigating the PD-L1 evaluation with CPS scoring in a series of selected cases of HNSCC from different Italian Pathology centers, using both LDTs and the CDx approved by FDA. The aim was to uniform and harmonize the assessment of the PD-L1 CPS score in clinical practice and explore a digital pathology workflow allowing second opinion consultation and the application of computational tools.

## 2. Materials and Methods

### 2.1. Cases

The organizing committee of this educational program asked 11 Italian labs to select one challenging HNSCC case/center, based on the assigned requests listed in Table 1. Histological samples were both surgical samples and biopsies. Cases of HNSCC were diagnosed according to the “WHO Classification of Head and Neck Tumors, 4th edition” (2017) [8]. Every center participated in the project with:One hematoxylin–eosin (H&E)-stained section;One LDT PD-L1 IHC stained with Dako clone 22C3 (Dako, Carpinteria, CA, USA) or Ventana clone SP263 (Ventana, Tucson, AZ, USA) and tested on the Dako or Ventana platforms, respectively;One blank section for IHC that was centralized at the Pathology Department of Fondazione IRCCS San Gerardo dei Tintori, Monza, Italy, where a pharmDx assay CDx (Dako) was performed.

All the samples were anonymized, and the study was carried out according to the principles of the Declaration of Helsinki. The study was approved by the local ethics committee (prot. 001716, S-PD-L1, 12 January 2021).

### 2.2. Digital Pathology

Slides were scanned using the PANNORAMIC MIDI II scanner (3DHISTECH, Budapest, Hungary) at 40× magnification. Subsequently, the digital slides were uploaded onto the digital platform DIPAP (DIgital PAthology Platform, Medica—Editoria e Diffusione Scientifica Srl, Milan, Italy) for remote consultation and expert board discussion.

### 2.3. PD-L1 CPS Scoring

The digital slides were made available to a board of 5 Italian pathologists (GD, AE, NF, MM, FP) with trained experience in the IHC evaluation of PD-L1 expression status [9], and they reevaluated the entire series, in blind, for a total of 110 scores. For the CPS scoring, it was considered the number of PD-L1 positive cells (both tumor and mononuclear inflammatory cells, including lymphocytes and macrophages) divided by the total number of viable tumor cells and multiplied by 100.

Key elements considered for the formulation of the score were:Presence of at least 100 viable tumor cells (adequacy criterion);Score range values from 0 to 100, with 100 as the maximum value;Evaluation of the staining at 20× magnification.

Tumor cells were considered positive in the presence of a convincing partial or complete linear membrane staining of any intensity. The inflammatory cells were considered positive in the presence of any distinguishable membrane and/or cytoplasmic staining, regardless of the inflammatory cell type or the specific staining location. The inflammatory cells considered in the score were lymphocytes and macrophages associated with the intratumoral and peritumoral stroma and included in the same microscopic field at 20× magnification. To determine the CPS score, the board was asked to use a complete scale of discrete values between 0 and 100. The PD-L1 CPS cutoffs relevant for clinical purposes in HNSCC were identified as CPS < 1,CPS ≥ 1,CPS ≥ 20. For each individual case, the PD-L1 CPS was assessed on the original LDT and subsequent CDx-stained slides. Results of the pathologists’ assessment were reported in an Excel spreadsheet (Microsoft, Redmond, WA, USA). The results collected were subsequently discussed by the board’s pathologists in a face-to-face meeting using whole slide images (Rome, May 2023) as a part of an educational program for pathologists from all over Italy. A definitive “consensus” was reached for the PD-L1 CPS on each case. All the pathologists attending the meeting were directly involved in the discussion. Afterwards, the following comparisons were carried out:LDTs PD-L1 (board consensus) vs. CDx PD-L1 (board consensus);LDTs PDL1 (board consensus) vs. LDTs PD-L1 (primary local diagnoses);CDx PD-L1 (board consensus) vs. LDTs PD-L1 (primary local diagnoses);

The following were considered:The technical concordance of the IHC stainings for PD-L1;The interpretative concordance of the two PD-L1 CPS scores.

### 2.4. Computational Analysis

All the PD-L1 digital slides were later analyzed using the open-source image analysis platform QuPath v0.4.4 [10]. For each digital slide, the color profile “Brightfield Hematoxylin-Diaminobenzidine (H-DAB)” was selected, and a semi-automatic color deconvolution was performed to separate hematoxylin from DAB channels. Subsequently, the following multi-step approach was adopted:Tissue was identified from the background using a thresholding technique based on image-specific parameters.Regions of interest (ROIs) for “Tumor” and “tumor + stroma” were identified and annotated manually. The stromal component was considered peritumoral when included in a microscopic field at 20× magnification, placing the tumor/peritumoral stroma interface at the center of the field.Automatic cell identification within the “tumor + stroma” annotations was achieved with the nuclear detector StarDist [11]. For this task, a custom script built for H&E images was used and modified to locate cell nuclei with a threshold of 0.2 and the surrounding cytoplasm based on nuclear expansion.


Finally, the distinction of “tumor” vs. “other” cells was performed by training an object classifier. Quantitative measurements of DAB intensity, including mean, median, standard deviation, and range, were extracted for these cell populations within different cell compartments (nuclear, cytoplasmic, and membrane). The obtained data were saved and transferred to Microsoft Excel for statistical analysis.

## 3. Statistical Analysis

Percentages of agreement and disagreement were obtained with respect to technical and interpretative evaluations. To assess the discrepancies among the three sets of evaluations, non-parametric tests for the comparison of median values, Kruskal–Wallis, and Wilcoxon were conducted, deeming values with *p*-values < 0.05 as statistically significant. Cohen’s kappa was employed to evaluate the consistency in classifying CPS scores as either above or below the threshold of 20. Regarding the image analysis study, for each pair of digital slides, the DAB intensity values (min, max, median, 1st quartile, and 3rd quartile) present at the levels of the two cellular subpopulations “tumor cells” and “other cells” forming part of the same annotation “tumor + stroma” were compared in pairs, and box-plot graphs were created. In cases where the box-plot graphs showed significant differences in terms of data distribution, the student’s *t* test was used to perform comparisons between DAB means of the “tumor cells” and “other cells” subpopulations (the significance level was set equal to α = 0.01), using the Bonferroni correction for multiple comparisons. The software used were Microsoft Excel 365 (Microsoft Corporation, Redmond, WA, USA), Pandas 2.2, and scikit-learn 1.4 (Python Software Foundation 3.10, Wilmington, DE, USA).

## 4. Results

### 4.1. Inter-Assay Variability

The PD-L1 CPS evaluated by the local center on LDT and by the board on LDT and CDx, along with technical and interpretative concordances/discordances highlighted by the board on LDT vs. CDx, are reported in Table 2 and Figure 1.

### 4.2. Technical and Interpretative Evaluations

Statistically significant differences were not found when comparing the three evaluations (*p* = 0.99, Figure 1). Major impact on the final assessment was noted when comparing batches with alternative stainings (LDT/board vs. CDx/board), with technical issues in five cases (45%, n° 2, 3, 7, 9, 11, LDT vs. CDx). Particularly, problems were mainly due to abnormal nuclear (case n° 2, Figure 2a) or cytoplasmic (case n° 3, Figure 2b) staining in the LDT assays (n = 2, 18%), whereas variability in terms of staining intensities, dirty background, and DAB droplets was recorded as more critical by board reviewers, alternatively affecting the LDT or CDx assays (cases n° 7, 9, and 11; n = 3, 27%; Figure 2c,d). Interpretative variability was seen in six cases (54%, n° 1, 2, 3, 7, 9, and 11; LDT/board vs. CDx/board) due to technical problems but mainly due to the particular challenges of the series (Figure 3, Table 1). When both variables were considered (staining/readers, LDT/local vs. CDx/board), differences were noted as still not reaching a statistically significant difference, as outlined by pairwise analysis (*p* = 0.72).

### 4.3. Impact of Variability on CPS Cutoffs

In some cases, the described technical issues and interpretative variabilities had an impact on the assignment of CPS above/below relevant cutoffs in HNSCC. In one case (n° 9, 9%), CDx led to an underestimation of the CPS, as compared to the LDT-based one, bringing the case below the cutoff of 20. Even if limited by the low number of cases available in this educational program/study, this resulted in minor differences in LDT/local vs. LDT/board (*k* = 0.82), with a slightly greater impact on LDT/board vs. CDx/board reproducibility (*k* = 0.63). Moreover, in three cases (n° 1, 2 and 3, 27%), both the LDT and CDx assays did not allow a sharp definition of the CPS value around the 1 (n° 2) and 20 (n° 1 and 3) cutoffs. An increase in CPS values from the LDT to the CDx assay was recorded in two cases (n° 7 and 11, 18%), reaching one of the two relevant cutoffs.

### 4.4. Computational Analysis

The application of the QuPath-based computational pipeline allowed a better visual evaluation of positive cells at the tumor–stroma interface, helping to understand the subtle discrepancies noted in some cases (Figure 4). The computational analysis of the intensity and distribution of DAB (tumor vs. non-tumor) in the LDT vs. CDx comparison did not show significant differences in nine cases (81.8%; n° 1, 2, 3, 4, 5, 6, 7, 8, and 10) (Appendix A). In the remaining two (18.2%; n° 9 and 11, Figure 5), an overall lower DAB intensity can be noted in the original LDT-based preparation (*p* < 0.01), potentially explaining the consequent underestimation of the CPS in case n° 11 (about 1 vs. 2 in CDx) and the paradoxical overestimation in case n° 9 (25 vs. 15 in CDx).

## 5. Discussion

In the report of this educational program, we focused on the evaluation of the expression of the predictive tumor response marker PD-L1 via the CPS score in a digital series of selected challenging cases of HNSCC. For each case, a board of expert pathologists evaluated an initial original staining performed with an LDT protocol and a second CDx staining. Subsequently, a discussion in a face-to-face meeting was carried out to discuss the variability linked to the methods. This study highlighted discrepancies both in terms of technical and interpretation factors. As per the technical part, both abnormal nuclear/cytoplasmic PD-L1 staining (18%) and intensity variability with dirty backgrounds (27%) were noted, suggesting a role for both pre-analytical and analytical phases. Many factors can influence the performances of the PD-L1 IHC, including cold ischemia time, fixation, thickness, and age of histological sections or FFPE samples [1]. The thickness of the histological sections must be 4–5 μm, and, if unstained, these must not be older than 2 months to avoid the possible loss of immunoreactivity [1]. In terms of stored material, it is worth remembering that samples older than 12 months should be avoided, as they may show reduction in PD-L1 expression [1,12]. Factors such as DAB droplets, solitary dotting and other spots, background staining, and edge artifacts could instead be connected to technical aspects of the method, causing possible interference with the interpretation [1]. Fixation, processing, incomplete removal of paraffin, and incomplete rinsing of reagents from the slide affect the presence of background staining; a comparison with the control can help perceive the levels of these artifacts [1]. Considering the LDT clones, three out of eight cases stained with 22C3 and two out of three cases stained with the clone SP263 were associated with a technical discordance or critical issues, compared to the CDx Dako 22C3 pharmDx assay. Despite the use of the same antibody, platform, and detection kit, different parameters may vary among laboratories using alternative protocols, for example, the type and duration of antigen retrieval, the dilution of the primary antibody, the incubation time, and the amplification [13]. In addition, the Ventana platform produces a staining with a more intense immunoreactivity but with a background tissue that often appears “burnt/crushed” and is not perfectly preserved, which often makes it difficult to distinguish the various cellular types; however, with the Dako platform, the immunoreactivity is apparently milder, but there is a gain in terms of the morphological definition of the various cell populations. When using an LDT, validation with a standard method remains crucial [1,14]. Studies using tissue microarrays (TMA) have demonstrated that Dako 22C3 pharmDx and Ventana SP263 assays present similar distributions of PD-L1 expression in the tissue sections, although the second assay may produce more false-positive results due to stronger and more widespread staining at the tumor cell level [9,15]. From the interpretative point of view, the presence of unspecific DAB droplets may represent a confounding factor, leading to an underestimation of the CPS, as compared to preparations with cleaner backgrounds (cases n° 7 and 9). Staining intensity can represent another source of CPS variability, with lower stainings being a potential cause of underestimation, due to the reduced perception by the human eye (case 11). Moreover, abnormal nuclear/cytoplasmic chromogen localization (cases n° 2 and 3) can have repercussions on the formulation of the PD-L1 CPS score as well, especially in cases fluctuating around relevant cutoffs of 1 or 20, hampering final assignments above/below these values. However, it is possible to note that in our case series, all the interpretative problems involved PD-L1 CPS values very close to the cutoffs relevant for clinical purposes (i.e., 1 and 20). Various results have been reported regarding the agreement between pathologists in the formulation of the PD-L1 CPS score in HNSCC cases. If it is associated with adequate training, the reproducibility of the score can be high, reaching an intraclass correlation coefficient (ICC) even ≥0.70 [9], ≥0.80 [16], or ≥0.90 and even using various different assays (22C3 pharmDx, SP263, and SP142) [15,17]. In terms of clinical impact, here, interpretation variability could have been relevant either for therapeutic (case n° 2, CPS score 1) or prognostic (case n° 7, CPS from 15 to 22) purposes, again stressing the potential impact of the different staining techniques around sensitive cutoffs (LDT/board vs. CDx/board *k* = 0.63). To address this evaluation heterogeneity, digital pathology can be a natural solution for the assessment of CPS scoring [18]. Here, we adopted the intrinsic capabilities of QuPath to perform subjective qualitative or semi-quantitative evaluation to establish eventual differences in terms of DAB distribution/intensity among the different PD-L1 preparations in order to understand technical/interpretative discrepancies observed by human eyes. The data revealed significant differences in both comparisons between the two cellular subpopulations only in two cases (n° 9 and 11), in which the board also noted discordances in staining at the level of visual evaluation. The dissociation between computational results and the technical/interpretative differences noted by pathologists in the remaining cases can be explained by multiple factors. In particular, human evaluation is affected by both visual and cognitive traps, which are also present in the expert pathologist, as they are not linked to experience but are intrinsic to the human being, but they are absent in the computational data derived from the image analysis, since they are, by their own nature, quantitative and objective [19,20]. Here, computational methods were used retrospectively to understand technical/interpretative heterogeneity, but in a routine clinical context, they can be fundamental for the quality assessment of the IHC preparations performed with different methods and can be applied directly by pathologists for diagnostic/predictive purposes. This may be a game changer in the context of PD-L1 evaluation, since in many countries, it is not necessary to use the FDA-approved CDx; however, it is important only to use a test that has been validated for the specific aim [21]. Furthermore, it can prove to be an essential method for the evaluation and management of histological preparations aimed at new tools connected to the world of digital pathology, including artificial intelligence (AI) algorithms, in which the quality of the analyzed data is greatly affected by pre-analytical and technical variables.

## 6. Conclusions

In the report of this educational program, we presented a frame of the current landscape of PD-L1 CPS scoring in HNSCC in our laboratories, underlining the importance of good preanalytical and analytical phases and also the crucial role of training, considering the clinical importance of this score in head and neck cancer. Digital pathology may serve as a facilitator to increase performances in PD-L1 scoring, both as a second opinion consultation platform in challenging cases and through the application of computational tools. The natural evolution would be the application of AI techniques to integrate this information to improve clinical decision-making and patient outcomes.

## Figures and Tables

**Figure 1 jcm-13-01240-f001:**
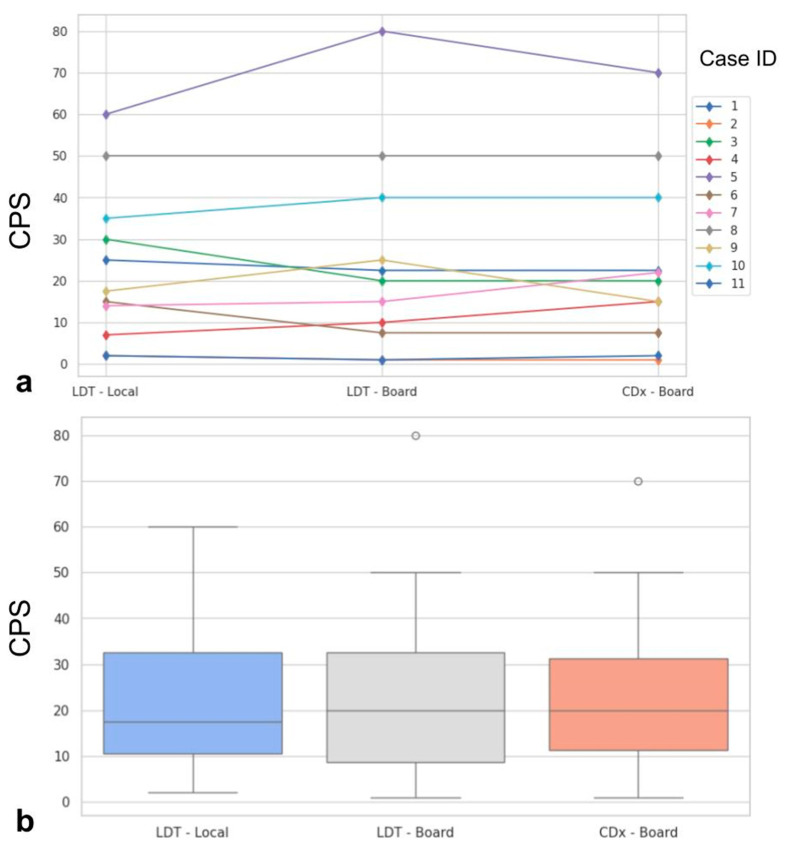
Spaghetti plot showing the case-by-case modifications of the PD-L1 CPS from LDT evaluated by local pathologist, LDT evaluated by the board, and CDx evaluated by the board (**a**). In (**b**), boxplots show the distribution of PD-L1 CPS in the three evaluation sets. ID: identifier; PD-L1: programmed death ligand 1; CPS: combined positive score; LDT: laboratory-developed test; CDx: companion diagnostic.

**Figure 2 jcm-13-01240-f002:**
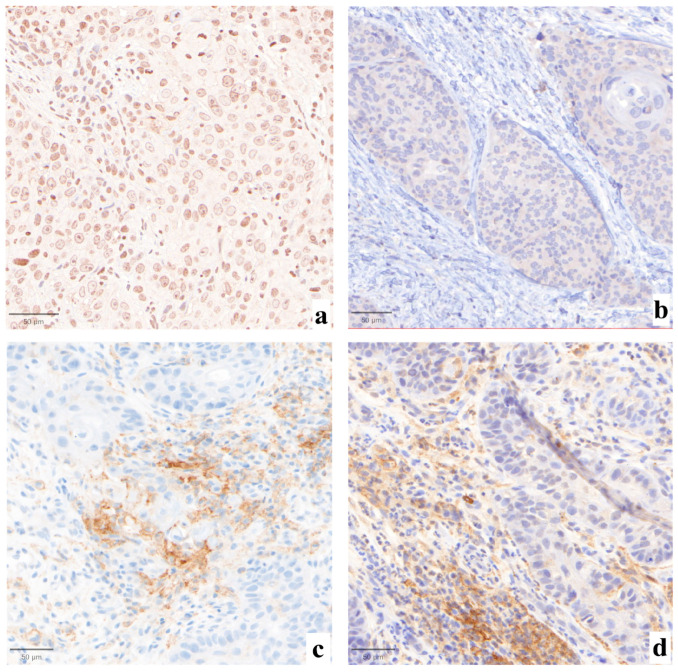
Abnormal nuclear ((**a**), 30×) or cytoplasmic ((**b**), 30×) staining with original PD-L1 LDT IHC in cases n° 2 and 3, respectively. Case n° 9 was affected by major technical issues, as per board judgment, showing a less intense but sharper membrane staining with the LDT assay ((**c**), 30×), as compared to the CDx-based preparation ((**d**), 30×).

**Figure 3 jcm-13-01240-f003:**
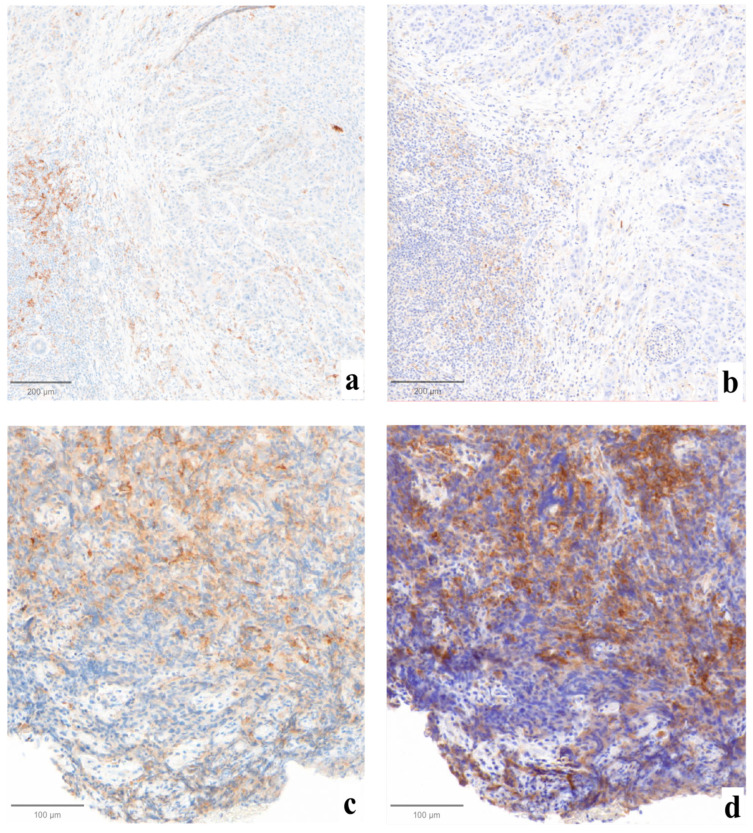
Examples of cases affected by interpretative variability due to technical factors. In case n° 7, sharper staining was obtained with LDT ((**a**), ×10), where fewer positive cells were counted, as compared to the more intense CDx-derived preparation ((**b**), ×10) that received a higher CPS. Case n°1 was affected by significant crush artifacts both on LDT-stained ((**c**), ×10) and CDx-stained ((**d**), ×10) tissues, complicating the single-cell localization of the positivity by pathologists.

**Figure 4 jcm-13-01240-f004:**
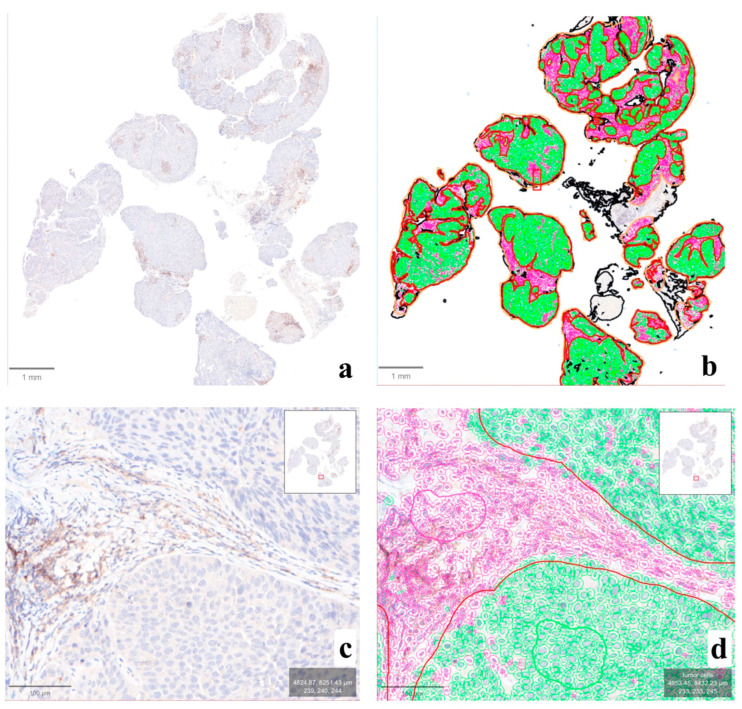
Output of the computational analysis performed on QuPath. In this case (n° 10), the original IHC slide ((**a**), ×1) was processed for the automatic detection of tumor (green) vs. non-tumor (violet) cells ((**b**), ×1), allowing the subsequent comparative analysis between compartments and assays. A closer look at the tumor–stromal interface ((**c**), ×10) demonstrates the impact of the evaluation of immune-positive cells in this region for the elaboration of CPS, easily computable through the application of the proposed QuPath pipeline ((**d**), ×10).

**Figure 5 jcm-13-01240-f005:**
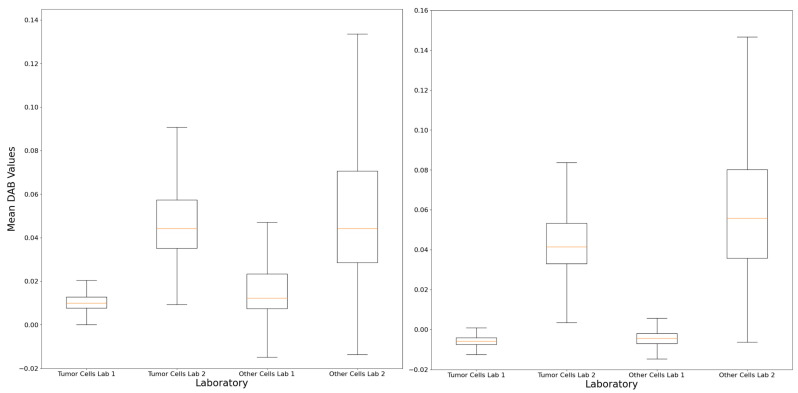
The computational analysis demonstrated differences in terms of DAB staining features within the tumor and non-tumor regions between the LDT and CDx preparations, as demonstrated by the boxplots of case n° 9 (**left**) and n° 11 (**right**), confirming the possible analytical reasons for the interpretation variability described. Lab 1: original local laboratory; Lab 2: reference laboratory.

**Table 1 jcm-13-01240-t001:** Challenges and assigned requirements for the study enrollment. LAB: laboratory.

LAB	Requirement Assigned	Sample Type	PD-L1 Clone
**1**	Crushing artifacts	Biopsy	22C3
**2**	Around CPS 1	Biopsy	22C3
**3**	Around CPS 20	Surgical	22C3
**4**	Fragmented sample	Biopsy	22C3
**5**	Great inflammatory component prevalence	Surgical	22C3
**6**	Lymph node metastasis	Surgical (lymph node metastasis)	22C3
**7**	Extensive necrosis	Surgical	SP263
**8**	Cell block 1	Biopsy	SP263
**9**	Cell block 2	Biopsy	22C3
**10**	Tumor component prevalence	Biopsy	22C3
**11**	Scant material	Biopsy	SP263

**Table 2 jcm-13-01240-t002:** PD-L1 CPS evaluation by local centers and the board on PD-L1 LDTs vs. CDx and technical/interpretative concordance, as per board judgment on the LDT vs. CDx PD-L1 IHC. Green: concordance. Pink: discordance. * “Around 20”: Case 3 underwent a long discussion between the members of the board, but they did not reach a definite final consensus on the CPS score considering the CPS 20 cutoff, both on the LDT and on the CDx stainings. “+”: In case 6, CDx staining was considered clearer than LDT staining but not influential on the final CPS score. ID: identifier; PD-L1: programmed death ligand 1; CPS: combined positive score; LDT: laboratory-developed test; CDx: companion diagnostic.

Case ID	CPS-LDTs	CPS-LDTs	CPS-CDx	Technical Concordance	Interpretative Concordance
	Local Evaluation	Board Evaluation	Board Evaluation		
**1**	25	20–25	20–25		
**2**	2	1	1		
**3**	30	around 20 *	around 20 *		
**4**	5–9	10	15		
**5**	60	80	70		
**6**	10–20	5–10	5–10	+	+
**7**	14	15	22		
**8**	50	50	50		
**9**	15–20	25	15		
**10**	35	40	40		
**11**	2	1	2		

## Data Availability

The data supporting the findings of this study are available from the corresponding author upon reasonable request.

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
