# Peer review of "Digital Pathology Applications for PD-L1 Scoring in Head and Neck Squamous Cell Carcinoma: A Challenging Series"

_jcm, 2024, doi:10.3390/jcm13051240_

Round 1
Reviewer 1 Report
Comments and Suggestions for Authors
The manuscript represents a study addressing the concordance between diagnostic and laboratory based assessment of PDL1 scoring.
The main issues are:
1. The small sample size (11) of the cohort preclude comprehensive assessment meaningful statistical correlation and do not support any of the conclusions.
2. The selection and application of computational programs are unclear and do not allow for meaningful interpretations of the results to clarify the clear explanation of inter-laboratory variabilities.
Reviewer 2 Report
Comments and Suggestions for Authors
This study aimed to compare the the diagnostic performances between local pathologists and a board of experts in the assessment of programmed death-ligand 1 (PD-L1) Combined Positive Scoring (CPS) on a challenging series of head and neck squamous cell carcinoma (HNSCC) through a digital pathology platform. The authors have reported that discordances were noted in 5 (45%) cases, 2 (18%) due to abnormal nuclear/cytoplasmic localization of the LDT PD-L1 stain and 3 (27%) due to variation in terms of intensity, dirty background and diaminobenzidine (DAB) droplets.
Specific comments
1. The number of samples used in this study was 11, which seems to be low from a statistical point of view. Was this number obtained by statistical methods?
2. From the statistical tests, it seems that it has not been used. Are the obtained percentages statistically significant or not?
3. The quality of the image 3 is not suitable. It is better to get a clearer figure.
4. What is the difference between this study and the study of references 26 and 29?
Round 2
Reviewer 2 Report
Comments and Suggestions for Authors
The authors addressed the all comments